# Microstructure and Properties of Magnesium Alloy Joints Bonded by Using Gallium with the Assistance of Ultrasound at Room Temperature

**DOI:** 10.3390/ma16216994

**Published:** 2023-10-31

**Authors:** Qiuyue Fang, Zuoxing Guo, Liang Zhao, Yuhua Liu

**Affiliations:** 1College of Materials Science and Engineering, Jilin University, Nanling Campus, Changchun 130025, China; fangqy21@mails.jlu.edu.cn (Q.F.); guozx@jlu.edu.cn (Z.G.); 2Key Laboratory of Automobile Materials, Ministry of Education, Jilin University, Changchun 130025, China

**Keywords:** room-temperature bonding, ultrasound, intermetallic compounds, gallium, magnesium alloy

## Abstract

Although magnesium alloys show potential as structural and functional materials, they are difficult to join using traditional welding methods because of their low melting points and active chemical properties. Their poor weldability impedes their universal application. Ultrasound-assisted transient liquid-phase bonding (U-TLP) is a novel method used for magnesium alloy bonding, but in almost all related studies, a heating device has been required, and the types of solders are limited. In this study, gallium was used as solder to bond AZ31 magnesium alloy with ultrasonic assistance at room temperature (without a heating device) due to the low melting temperature of gallium and its compatibility with other metals when forming intermetallic compounds (IMCs). The variations in the products, microstructure, fracture characteristics, and shear strength of the joints were investigated. A reliable joint composed of IMCs (Mg_2_Ga_5_, H-MgGa_2_, and Mg_2_Ga) and a eutectic structure was obtained after an ultrasonic duration of 3 s. Significantly, the plasticity of the joint was improved due to ultrasonic effects, which included the accelerated element diffusion process, the refinement of grains to nanometer particles, and the homogenization of organization. Thus, the highest shear strength of 14.65 MPa at 4 s was obtained, with obvious cleavage fracture characteristics in the region of the IMCs.

## 1. Introduction

Magnesium alloys have attracted attention from numerous actors in the aerospace, automobile, military, and medical material industries because of their high specific strength, conductivity, machinability, magnetic shielding, and biocompatibility [1,2]. Many welding methods, such as arc welding, brazing, friction stir welding, laser welding, and resistance spot welding, have been applied in order to bond magnesium alloys and obtain high-quality joints [3]. Nevertheless, problems such as cracks, holes, and coarse grains are still observed, and these affect the reliability of such joints. In most welding methods, fluxes with complex components are used to eliminate oxide film and inclusions on the interfaces of the base and filler metals. Additionally, a shielding gas or vacuum environment is used to protect the joints from oxidation during welding. These methods require complicated devices. In addition, the toxic gases generated are damaging to human health and the environment [4]. Therefore, to reduce costs, save energy, and protect air quality, it is vital to develop fast welding methods that do not require flux and that operate at low temperatures.

In recent years, ultrasound-assisted transient liquid-phase bonding (U-TLP) has attracted much attention for joining magnesium alloys because of the “cavitation” and “streaming” effects of ultrasound; this method can be used to remove oxide film, improve wettability, accelerate diffusion, refine grains, and create a well-organized distribution. Lai et al. bonded ME20M magnesium alloy using U-TLP with pure Zn as an interlayer. They used induction heating circuits to maintain the bonding temperature at 370 ± 5 °C. A joint composed of α-Mg solid solution was obtained after 120 s, and it had a shear strength of 106.4 MPa with a fracture extending from the base metal [5]. Additionally, Lai et al. used brass as an interlayer when bonding ME20M magnesium alloy in air to study the mechanism of oxidation film removal through ultrasound. They produced joints with a shear strength of 105 MPa after 90 s at 460 °C and proposed four steps for oxide removal [6]. While using a Zn interlayer to bond ME20M magnesium alloy, Xie et al. carried out the bonding process at 490 °C. A novel method of secondary ultrasonic treatment was used in their study, and a joint composed of a solid solution with excellent quality was obtained. The highest shear strength reached 109.4 MPa. They also found that the amount of IMCs in the joints gradually decreased and finally disappeared with the increase in the bonding temperature [7]. Subsequently, Xu et al. bonded AZ31 magnesium alloy while using a Zn interlayer to study the effects of temperature and ultrasonic duration on the structure and mechanical performance of joints [8,9]. They found that higher temperatures helped with the formation of a eutectic structure, and longer ultrasonic durations and higher pressures were also helpful. Two highlights of their study were the lower bonding temperatures (360–380 °C) and extremely short bonding times (0.1 s). However, the shear strength of the joints obtained was relatively lower—about 30 MPa for joints composed of a eutectic structure and IMCs and 40 MPa for eutectic joints. Wang et al. chose a Ag-28Cu foil interlayer to join MB8 magnesium alloy at 480 °C and studied the effect of ultrasonic duration on the quality of the joints [10]. They found that with a prolonged ultrasonic duration, the ternary eutectic liquid phase of Mg-Ag-Cu increased, and the weld gradually closed. The joints consisted of AgMg_3_, CuMg_2_, and α-Mg solid solution, and the fracture morphology was characterized by brittle fractures when considering short durations, but presented ductile characteristics with longer exposure to ultrasound. Li et al. chose Ni as an interlayer for bonding Mg-Mn-Ce alloy at 520 °C to explore the effect of ultrasonic vibration time on the microstructure and mechanical properties of joints [11]. With incremental increases in time up to 30 s, the shear strength increased from 19 MPa to 92 MPa, which might have resulted from a decrease in the amount of IMCs. However, when the ultrasonic duration was increased to 35 s, the shear strength rapidly decreased because of an unfilled region that was observed. Zhang et al. bonded magnesium alloy while using Al as interlayer at 450 °C; they also found that the amount of IMCs decreased with the prolongation of the ultrasonic duration, and the weld gradually closed. The highest shear strength of a joint was 80.26 MPa after 30 s [12].

In these studies, a furnace and induction heating devices were needed to heat samples to at least the eutectic point or higher (about 360–520 °C) to produce a “cavitation” effect, which requires a liquid environment [13,14]. IMCs are not expected to exist in such joints because their hard and brittle characteristics are always harmful to the microstructure and mechanical properties of joints [15,16]. Through some approaches, such as raising the joining temperature [11], prolonging the duration of ultrasound [9], using secondary ultrasound [17], and water cooling [18], IMCs can be decreased and eliminated. U-TLP combines the advantages of being flux-free, requiring no serious gases, and having a short joining time, making it of vital importance in the development of magnesium welding techniques. However, the requirement for a heat input still necessitates heating devices and security guarantees for operators. Additionally, high joining temperatures intensify the oxidation process in magnesium, which might have an impact on the application of magnesium alloys in some circumstances.

In this study, to further simplify the welding device and reduce the joining temperature, we joined a magnesium alloy by using gallium (Ga) as the solder with the assistance of ultrasonic vibration. This is the first time that a magnesium alloy has been bonded by using U-TLP at room temperature and under a normal atmosphere without an external input of heat. The influence of the ultrasonic duration on the microstructure and mechanical properties of the joints was also studied. Based on the merits of dimensional accuracy and the original performance of the base materials due to low-temperature bonding [19,20], this study has meaningful value for broad applications of magnesium alloys in the future and the exploration of low-temperature bonding for other materials.

## 2. Materials and Methods

### 2.1. Materials and Devices

AZ31 magnesium alloy (97% Mg, 3% Al, and 1% Zn) was used as a base metal in this study, and it was cut into 20 mm × 15 mm × 4 mm and 15 mm × 10 mm × 4 mm bulk samples. Then, these bulk samples were ground with abrasive paper to #800 and carefully cleaned. Pure gallium (Ga), which has a melting temperature as low as 29.8 °C and has the ability to react with almost all metals at room temperature [21], was used as the solder. The ultrasound welding system was custom-made and manufactured by Hangzhou Successful Ultrasound Equipment Co., Ltd. (Hangzhou, China). As shown in Figure 1c, it mainly consisted of an ultrasonic generator, a pressure system, and a time control system. The frequency of the ultrasonic generator was 20 kHz, and the power was within 2000 W. The suggested application temperature for this system was 0–40 °C, and the power automatically varied during the welding process.

### 2.2. Experimental Process

First, solid pure gallium was heated to 50 °C, and the temperature was maintained for 10 min to guarantee it that it fully melted until liquid gallium was obtained. Then, this liquid metal was uniformly painted on one side of the base metal, as shown in Figure 1a, with a thickness of about 1–2 μm. Subsequently, two bulk samples of different sizes were positioned so that they were touching each other, and they were fixed in the welding mold. The larger sample was completely fixed, and the upper sample could move in one direction, thus limiting the samples’ movement as much as possible. After that, the whole mold was placed on the ultrasound generator, as shown in Figure 1b. The amplitude was set to 40 μm, and the pressure imposed on the samples was 13.3 MPa. After the pressure stabilized, the ultrasound was started, and it continued for 2 s, 3 s, 4 s, 6 s, 9 s, or 12 s. After joining, the samples were cut from the center with a wire-cutting machine to obtain cross-section samples. The original cross-section samples were ground with abrasive paper from #600 to #7000 and then polished using velvet and a diamond polishing agent with a particle size of 1 μm, followed by washing with ultrapure water. Then, the samples were positioned for a shear test, with the larger base metal being fixed and the load being imposed on the smaller base metal, as shown in Figure 1c. Then, the smaller shear plane was used for a phase analysis and observation of the fracture morphology. The shear strength of the joint was measured using mechanical testing and simulation (MTS) with a constant pressing speed of 0.2 mm/min at room temperature. The average shear strength value of three samples was regarded as the final shear strength for each experimental parameter.

### 2.3. Characterization

Scanning electronic microscopes (SEM, VEGA3 XMU, TESCAN, Brno, Czech Republic and JSM-7900 F, JEOL, Tokyo, Japan) equipped with an energy-dispersive spectrometer (EDS, Aztec Oxford, London, UK) were used to observe and analyze the weld microstructure and fracture morphology. The elemental distributions of the joints were analyzed using the EDS. The phases of the joints were identified using the X-ray diffraction (XRD, D/Max 2500 PC, Rigaku, Japan) method.

## 3. Results and Discussion

### 3.1. Microstructure of Joint

In order to reduce the influence of diffraction peaks from the base metal and to ensure the accuracy of phase determination, glancing incidence technology was used to scan the shear surface after the shear test. Figure 2 shows the XRD results of samples that were treated with different ultrasonic durations. Through a comparison with the phase diagram of binary Ga-Mg alloy [22], it was found that the joints were mainly composed of Mg_2_Ga_5_, H-MgGa_2_, and Mg_2_Ga and a eutectic structure (Mg_0.813_Ga_0.187_). According to the results of first-principles calculations by Liu L., et al., the formation enthalpy of H-MgGa_2_ was −0.144 eV/atom and that of Mg_2_Ga_5_ was −0.140 eV/atom; Mg_2_Ga_5_ had a tetrahedral crystal structure [22]. 

In the sample with an ultrasonic duration of 3 s, the diffraction peak of H-MgGa_2_ at (0 1 2) was the highest, and no peak of Ga appeared, which implied that the solder was consumed at 3 s. However, at 4 s, the peak of Mg_2_Ga_5_ at (0 0 2) was the highest. At 6 s, the peak of H-MgGa_2_ at (0 1 2) was the highest again, and the peak of Mg_2_Ga_5_ at (0 0 2) declined, which suggested that there was an increase in H-MgGa_2_ after an ultrasonic duration of 6 s. At 9 s, the peak of H-MgGa_2_ declined and that of Mg_2_Ga_5_ became the highest. The variations in the XRD peaks suggested that the compositions of the IMCs underwent some changes with increases in the ultrasonic duration. It is also noteworthy that the shape of the diffraction patterns changed at different times, indicating the existence of a crystallographic texture. 

Figure 3 shows the typical structure of a joint at 4 s. The weld consisted of a white layer in the center named L_1_ (layer 1) and a gray layer named L_2_ (layer 2), which was the result of mutual diffusion between Ga and Mg atoms. In the photographs of the elemental distribution in Figure 3d,e, it is clear that the Ga atoms were mainly distributed in L_1_, but there were fewer in L_2_, while the opposite was true for Mg atoms. This implied that L_1_ was mainly composed of Ga-rich phases, while L_2_ was more likely to consist of Mg-rich phases. Figure 3b was the amplified image of region A, which shows the typical unilateral structure of L_1_. The EDS results of point 1 to point 5 in Figure 3 werelisted in Table 1. It was noteworthy that L_1_ could be divided into three layers according to the different morphologies of the compounds; the first layer was the central area of the weld. There were few cavities, and the grains were nondirectional. Table 1 shows that the content of Ga was 60% and that of Mg was 40%. Based on the XRD results, it was determined that the region was composed of a mixture of Mg_2_Ga_5_, H-MgGa_2_, and some Mg_2_Ga. The second layer consisted of grains that presented a direction parallel to the direction of pressure. This morphology was probably due to the formation and growth of H-MgGa_2_ in this region, which had strong anisotropy [22]. Combined with the diffraction pattern, it could be considered that H-MgGa_2_ grew rapidly in the crystal plane (1 1 0) during the process of solidification. The third layer was very thin and dense, and it served as a transition layer between L_1_ and L_2_. The atomic percentages of Ga and Mg in this region were similar, indicating that it was composed of H-MgGa_2_ and Mg_2_Ga. Figure 3c was the amplified image of region B in Figure 3a. It shows the main structure of the gray layer L_2_, which had visible grains and grain boundaries. The Ga content accounted for 84.6%, indicating that it might have had a eutectic structure (Mg_0.813_Ga_0.187_). Figure 3f shows the honeycomb-shaped structure observed in L1. The EDS results indicated that the ratio of Mg to Ga was 2:1, so this was determined to be Mg_2_Ga. The existence of Mg_2_Ga in L_1_ suggested that in some regions, a large number of Mg atoms rapidly diffused over a long distance during the ultrasonic process. As a result, the formation of Mg-rich phases was achieved in the central layer of the weld.

The formation of a joint involves a process of diffusion between Ga and Mg to form IMCs. Similar to the low-temperature joining of copper [23,24,25,26], the formation of IMCs makes the joining of magnesium alloys successful at low temperatures. Before applying ultrasound, wetting had already occurred between the solder and base metals due to the affinity between them. In rapid diffusion channels, such as grain boundaries and dislocations, the mutual diffusion of Ga and Mg occurred, resulting in the formation of a very thin layer of Mg_2_Ga_5_ on one side of the Mg. Subsequently, Ga gradually became semisolid or solid before the pressure stabilized, which was due to the temperature reduction when touching the steel mold. Once the ultrasound began, due to the combined effects of friction, sonic pressure, and external load, the temperature easily rose above the melting point of Ga, and the Ga then began to melt and became liquid in the regions in which the melting conditions were met. It is certain that melting began at the interface of the solder and base metal because the difference in the atomic arrangements of the two different materials at the interface caused it to be in a high-energy state, so atoms could be more easily activated, and the chemical bonds between atoms could be more easily broken. This was also proven by the residual Ga at the central location of the weld after short ultrasonic durations. In liquid Ga, interatomic bonding became weaker, and a high concentration of vacancies was generated due to the “cavitation” effect of ultrasound over a short period of time [27], thereby promoting the diffusion of atoms. On the other hand, the “streaming” effect generated by ultrasound in liquid Ga would push the liquid to continuously flow toward both sides of the base metals. When the ultrasonic treatment was stopped, the temperature rapidly decreased, solidification occurred, and IMCs with fine crystal grains formed. When the “cavitation” effect of ultrasound occurred in liquid Ga, the explosion of cavitation bubbles caused local high-energy and high-temperature spots [14,28]; this was also the source of the energy and heat of the formation of the eutectic liquid. Because the conditions for reaching the eutectic point and attaining a eutectic composition at the same time were more easily met in the area near the base material, a eutectic liquid phase more easily formed there. After the ultrasound was stopped, a eutectic reaction (L (81.3 Mg, 18.7 Ga, at %) → Mg_5_Ga_2_ + α-Mg) occurred, thus forming a eutectic structure: Mg_0.813_Ga_0.187_. This result is similar to that obtained by Xu et al. when using U-TLP to bond AZ31B magnesium alloy [8]. A joint composed of a eutectic structure and IMCs was also obtained in their study. However, the eutectic structure was mixed with IMCs because they set the bonding temperature above the eutectic point, which caused the formation of a eutectic liquid phase from the beginning. However, in this study, the generation of a eutectic liquid phase did not rely on an external heat source; rather, it was a secondary liquid phase that was derived from the ultrasonic effect on liquid Ga.

### 3.2. Effect of Ultrasonic Time on Joint Structure

For joints obtained with different ultrasonic durations, the width of the weld slightly varied in the range of 18 ± 3 μm, so the variations in width are not discussed here. Rather, the microstructure of the weld is carefully discussed. Figure 4a–d show amplified images of the microstructure of the unilateral welds for ultrasonic durations of 3 s, 4 s, 6 s, and 9 s, respectively. It was clear that the microstructure of the welds changed with the increase in the ultrasonic duration.

When the ultrasonic duration was less than 3 s, Ga could not completely react, so there was a large amount of residue in the center of the weld. Under this condition, the connection between the base materials almost entirely depended on the solidification of Ga, but an effective connection was not achieved. During the grinding process as part of sample preparation, the Ga melted due to the increase in temperature caused by friction, thus causing the joint to slip open easily. Therefore, it was impossible to prepare a cross-section sample when the ultrasonic duration was less than 3 s. When the ultrasonic duration was prolonged to 3 s, the Ga almost completely melted and reacted. As a result, a joint mainly composed of IMCs was initially obtained. However, there were some cavities and a small amount of residual Ga in the area local to the weld, as shown in Figure 4a. Figure 4b shows the microstructure of the weld after 4 s. As discussed with reference to the XRD results, more Mg_2_Ga_5_ formed at 4 s. The crystals of Mg_2_Ga_5_ are smaller than those of H-MgGa_2_ [22], so fewer and smaller cavities emerged. In some regions, the columnar morphology of H-MgGa_2_ was observed.

When the ultrasonic duration was prolonged to 6 s, the amount of H-MgGa_2_ increased, as shown by the XRD patterns. As shown in Figure 4c, the columnar morphology of H-MgGa_2_ was discovered in more areas and became clearer, while the central grains of IMCs existed in elliptical and spherical shapes. There were obvious micro-seams between the columnar layers and the gathered bulks. When ultrasound was applied for 9 s, as shown in Figure 4d, the columnar morphology and large blocks no longer appeared in the joint. The voids and porosity also disappeared, and the structure of the entire joint was refined and homogenized. 

It was clear that with short durations, the energy provided by the ultrasound was not even in the different regions; thus, the distributions of the compounds and structures were different. With a duration of 6 s, the rate of growth was obviously faster than that of nucleation, which resulted in large grains. With longer ultrasonic durations, such as 9 s, generally, “cavitation” occurred everywhere in the liquid, thus providing an environment with a higher and more even temperature, which lasted longer and increased the generation of more crystal nuclei. Additionally, the IMCs that formed before was more likely to be fragmented and the components became more uniform at different regions under the action of adequate ultrasonic vibration, so more nucleation points formed [29,30]. Finally, the thermal conductivity of the Mg base metal implied a rapid decrease in temperature when the welding finished. As a result, the nucleation rate was larger and exceeded the growth rate in comparison with that at a duration of 6 s. These reasons accounted for the refinement of the IMCs and the homogenization of the structure of the joint at 9 s. 

However, excessive vibration also caused micro-cracks in the joints, as shown in Figure 4d. These might have been caused by the thermal stress generated in the process of rapid solidification. These micro-cracks grew and converged into continuous cracks when the ultrasonic duration was prolonged to 12 s, leading to deterioration in the structure and performance of the joint. An unfilled region was also observed at 12 s, and this was attributed to the massive squeezing of the liquid during the welding period. Changes in the structure of a joint affect its fracture characteristics and mechanical properties.

### 3.3. Fracture Locations

Figure 5a shows the phenomenon of the tearing of the top-side base metal after the shear test. This resulted from the local plastic deformation caused by ultrasonic vibration. Xie et al. suggested that plastic mismatch happens between metals and oxide film, which contributes to the breakage of oxide films; then, fresh metal and the filler touch, atoms diffuse, and a eutectic liquid forms [7]. Additionally, Somekawa investigated the superplasticity of magnesium under the action of ultrasound [27]. These studies indicate the possibility that local plastic deformation occurs at the interface during ultrasound. However, the specific location of the fracture was difficult to visually distinguish because of the rheological phenomenon caused by large-scale tearing. From the elemental distribution diagrams, it was determined that fractures mainly occurred in the IMC layers with more Ga, namely, the areas in which Mg_2_Ga_5_ and H-MgGa_2_ were aggregated, with some fractures occurring in the eutectic area. Figure 5b shows an SEM image of the fracture location in a joint with a shear strength of only 4 MPa. The fractures mainly occurred in the central Mg_2_Ga_5_-rich region, which was where cavities were distributed, or at the interface between the Mg_2_Ga_5_-rich layer and the columnar H-MgGa_2_ layer, which resulted in a relatively flat fracture morphology. When fractures penetrated more H-MgGa_2_-rich and eutectic areas, as shown in Figure 5c, tearing also occurred. There was an extremely rough fracture plane because of the wave-like fracture path. In Figure 5d, which shows the local rheological phenomenon, it can be seen that both the IMC layer near the base metal and the eutectic layer experienced tearing. This implied that the IMCs in these regions presented ductile characteristics. As shown in Figure 5f, the interfaces between regions with different crystal structures, such as the columnar layer and eutectic layer, were easily attacked. The amplified view of the fracture location in Figure 5e shows that cracks were still distributed in the IMC layers but stopped before they propagated to the eutectic layer. This phenomenon occurred in almost all fracture samples. Therefore, it could be hypothesized that the cracks could pass through the central IMC layer more easily because of the distribution of cavities. Additionally, the interfaces between two different layers were also options for cracks. However, the existence of the columnar and eutectic layers was instrumental in resisting crack propagation.

### 3.4. Shear Strength of Joint

Figure 6 shows the variations in the average shear strength of the joints with different ultrasonic durations. Each value is the average recorded for three samples with the same experimental parameters. Before the shear test, the edges of each sample were cleaned to eliminate any solder that squeezed out during welding to ensure the accuracy of the test. The shear strength of the joints increased from 2 s to 4 s and then decreased from 4 s to 12 s. When ultrasound was applied for 2 s, the average shear strength of the joint was only 2 MPa as a result of the unreacted Ga. Fractures almost always occurred in the residual Ga, making the strength of Ga in different states highly variable; finally, a very low average value was obtained. When the ultrasonic duration increased from 2 s to 3 s, the Ga was consumed, and the amounts of IMCs increased. Fractures mainly occurred in the regions composed of Mg_2_Ga_5_ and H-MgGa_2_, and a few occurred in the eutectic layer. Therefore, the average shear strength of the joints increased to 10.06 MPa.

When the ultrasonic duration was prolonged to 4 s, the average shear strength reached 14.65 MPa. As demonstrated by the discussion of the microstructure of the weld, more IMCs—especially Mg_2_Ga_5_—were formed, and the grains were denser and finer than those obtained with a duration of 3 s. Fractures mainly occurred in the Mg_2_Ga_5_ and H-MgGa_2_ columnar layers, with some passing through the eutectic layer, forming a wave-like propagation path, as shown in Figure 5c. When fractures passed through the H-MgGa_2_ columnar layer, a high shear strength was obtained because more energy is required to break atomic bonding within crystals than among crystals. 

When the ultrasonic duration was increased to 6 s, the connection loosened due to the growth of the grains of IMCs. Under this condition, cracks easily propagated in the gaps between IMCs, which counterbalanced most of the effect of the stronger H-MgGa_2_ columnar layer, causing the shear strength of the joint to decrease. When the ultrasonic duration exceeded 9 s, although the refinement and homogenization of the weld made the shear strength of the joint more stable, the shear strength further decreased. This was because micro-cracks were generated in the areas containing IMCs, as shown in Figure 4d. When the ultrasonic duration was prolonged to 12 s, micro-cracks gradually grew, and unfilled regions appeared, which resulted in a decrease in the shear strength of the joint.

### 3.5. Fracture Morphology

Figure 7 shows the fracture morphologies with different ultrasonic durations. As shown in Figure 7a, there were signs of cleavage in the fracture plane of the residual Ga at 2 s, which suggested that a cleavage fracture occurred in the solid Ga. The remaining areas demonstrated a morphology caused by the sliding of the residual Ga after melting during the shear process. Therefore, the fracture mode at this time was mainly that of sliding, and a cleavage fracture occurred in the residual Ga region. Figure 7b shows the typical fracture morphology of the joint with an ultrasonic duration of 3 s. Clear crystals were observed in the region of H-MgGa_2_; it was clear that the crystal size of H-MgGa_2_ was larger, and the connections between Mg_2_Ga_5_ grains were loose. The fracture mode was mainly that of the sliding of Mg_2_Ga_5_ grains and intergranular fracture in the region dominated by H-MgGa_2_, and a small amount of exposed eutectic structure was visible. This morphology implied that the fracture propagated in the Ma_2_Ga_5_ regions and sometimes passed through the H-MgGa_2_ and eutectic regions, as shown in Figure 5a. For an ultrasonic duration of 4 s, as shown in Figure 7c,d, a large area of cleavage appeared in the IMC region. A shear surface formed due to the aggregation of fine IMCs, and this presented radial shear patterns, which are typical characteristics of brittle fractures. In Figure 7c, it can be seen that several crystal orientations coordinated during the process of fracture, which occurred in the Ma_2_Ga_5_-rich region due to the low anisotropy of Mg_2_Ga_5_. However, the morphology shown in Figure 7d suggests that fractures passed the H-MgGa_2_ columnar layer, and these were transgranular fractures. When the ultrasonic duration extended to 6 s and 9 s, long shear ridges of IMCs with finer and denser grains and some plastic characteristics were observed. It seemed that the grain size was gradually refined as the ultrasonic duration was increased, thus improving the plasticity of the joint. As a result, the joint gradually underwent cleavage fractures with some ductile characteristics, which was consistent with the changes in the weld microstructure.

## 4. Conclusions

(1)AZ31 Mg alloys were successfully bonded using pure Ga at room temperature using ultrasound. The connection of the joints mainly relied on the reaction of Ga and Mg to generate IMCs (Mg_2_Ga_5_, H-MgGa_2_, and Mg_2_Ga) and a eutectic structure.(2)As the Ga solder was gradually consumed, a reliable joint could be obtained after an ultrasonic duration of 3 s. IMCs grew, refined, and homogenized with an ultrasonic duration that was further prolonged to 9 s. However, cracks and unfilled regions were observed with excessive ultrasonic durations.(3)The shear strength of the joints increased with durations from 2 s to 4 s and then steadily decreased from 4 s to 12 s, and the highest shear strength of 14.65 MPa was obtained after ultrasonic treatment for 4 s. Brittle fractures occurred, and the morphology expressed the typical characteristics of cleavage fracture and some plastic characteristics with the extension of the ultrasonic duration.(4)We developed a simple and fast bonding method for the low-temperature joining of magnesium alloys. Low-temperature bonding contributes to energy savings and allows the accuracy of the size and original properties of the base materials to be retained, which is useful for the bonding of components and parts that require high accuracy but a relatively low load.

## Figures and Tables

**Figure 1 materials-16-06994-f001:**
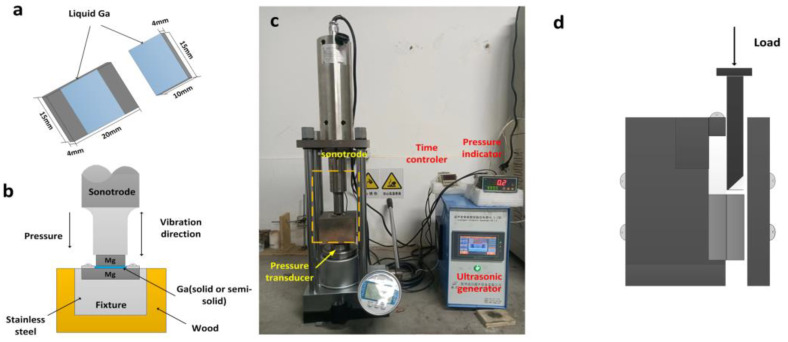
Schematic diagram of experimental process: (**a**) Mg base metals after painting liquid Ga; (**b**) sketch of joining configuration; (**c**) schematic diagram of ultrasonic system; (**d**) cross-sectional diagram of shear test.

**Figure 2 materials-16-06994-f002:**
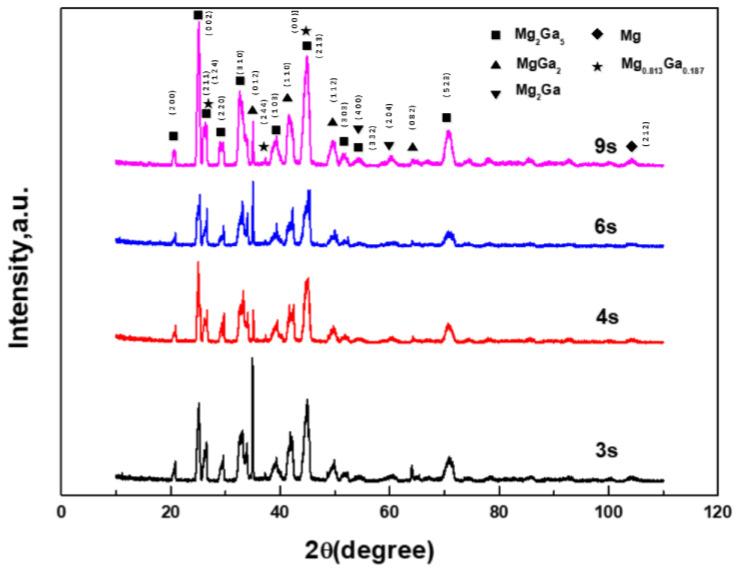
X-ray diffraction pattern of the shear planes at different ultrasonic time.

**Figure 3 materials-16-06994-f003:**
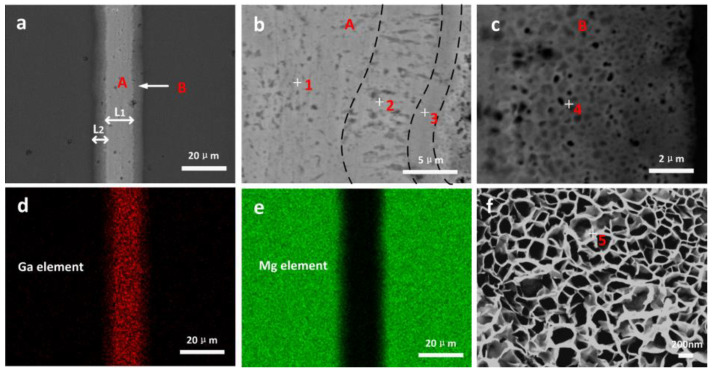
Structure of joint after 4 s ultrasonic treatment: (**a**) the whole weld; (**b**) amplified graph of region A; (**c**) amplified graph of region B; (**d**) the distribution diagram of Ga element; (**e**) the distribution diagram of Mg element; (**f**) amplified diagram of Mg_2_Ga existing in L_1_.

**Figure 4 materials-16-06994-f004:**
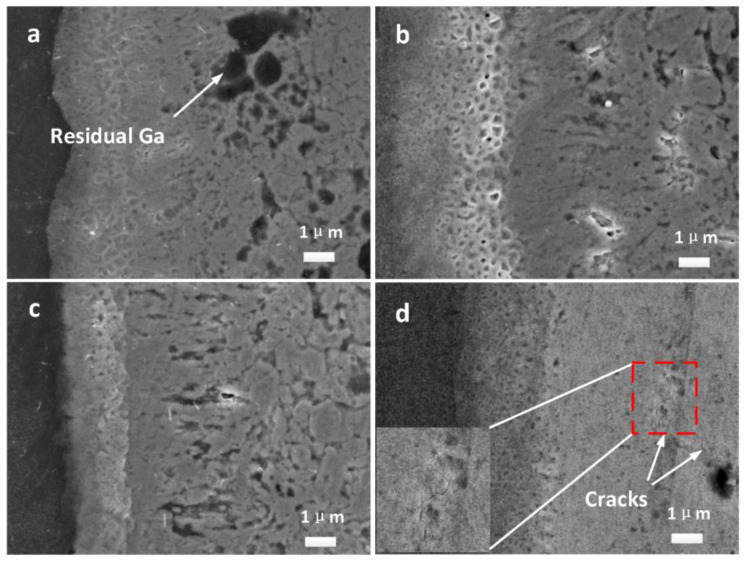
Amplified diagram of welds obtained at (**a**) 3 s; (**b**) 4 s; (**c**) 6 s; (**d**) 9 s, respectively.

**Figure 5 materials-16-06994-f005:**
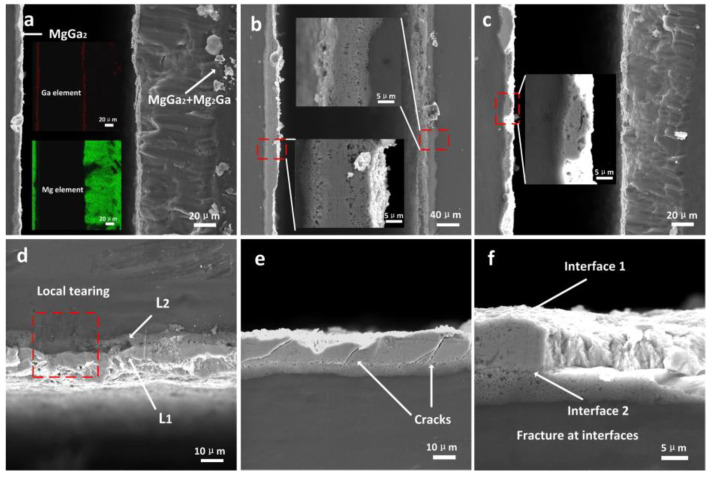
Diagram of fracture locations: (**a**) cracks propagate in the IMCs layer and product tearing, (**b**) cracks gently spread through central IMCs; (**c**) wave-like path of cracks that propagate in different layers and tearing phenomenon; (**d**) magnified diagram of the region of local tearing; (**e**) the region of cracks remaining in IMCs layer; (**f**) fracture passes through different interfaces.

**Figure 6 materials-16-06994-f006:**
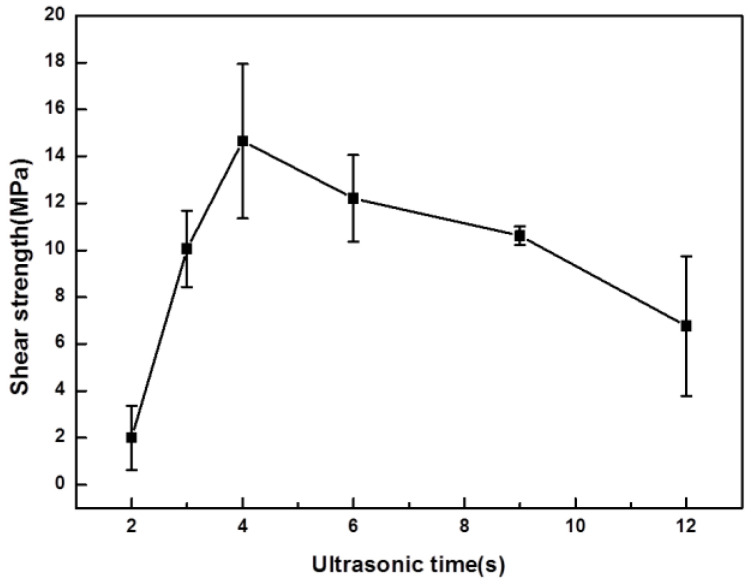
Results of average shear strength of joints for different ultrasonic durations.

**Figure 7 materials-16-06994-f007:**
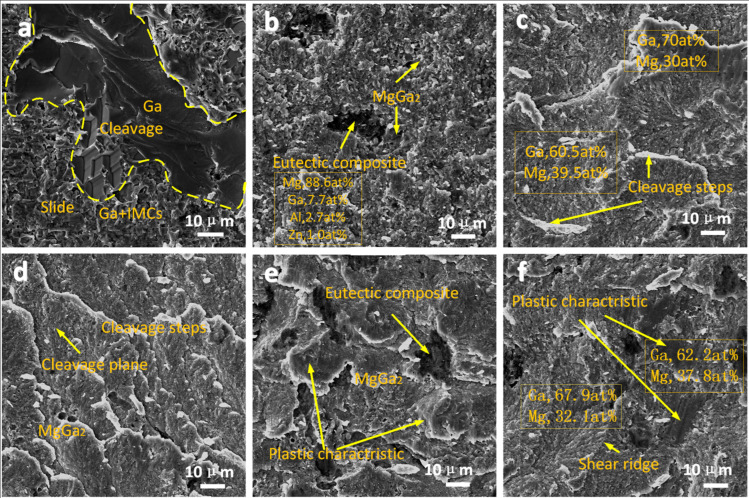
Diagrams of fracture morphology for different ultrasound durations: (**a**) 2 s, (**b**) 3 s, (**c**) 4 s, Mg_2_Ga_5_-rich region and (**d**) 4 s, H-MgGa_2_-rich region, (**e**) 6 s, and (**f**) 9 s.

**Table 1 materials-16-06994-t001:** Chemical composition of different regions in Figure 3.

Region	Mg (at%)	Ga (at%)	Possible Phase
1	40.0	60.0	MgGa_2_, Mg_2_Ga_5_, Mg_2_Ga
2	38.1	61.9	MgGa_2_
3	46.2	53.8	MgGa_2_, Mg_2_Ga
4	84.6	15.4	Mg_0.813_Ga_0.187_, Mg_2_Ga
5	66.4	33.6	Mg_2_Ga

## Data Availability

No data was used for the research described in the article.

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
