# Peer review of "Microstructure and Properties of Magnesium Alloy Joints Bonded by Using Gallium with the Assistance of Ultrasound at Room Temperature"

_materials, 2023, doi:10.3390/ma16216994_

Round 1

Reviewer 1 Report

Comments and Suggestions for Authors

Comments to the manuscript by Qiuyue Fang, Zuoxing Guo, Liang Zhao, Yuhua Liu “Microstructure and properties of magnesium alloy joints bonded by gallium under the assistance of ultrasound at room temperature”

The paper is devoted to the gallium usage as a solder to bond the AZ31 magnesium alloy under ultrasonic assistance at room temperature.

I have the following comments to the paper.

1.      Diffraction patterns on the figure 2 are not indexed.

2.      Pure Ga has the only peak on the black pattern. That peak coincides with one of MgGa2. It is not clear why it is interpreted as pure Ga peak instead of crystallographic texture influence on the peak of MgGa2.

3.      Intensity of the second peak of Mg2Ga5 is different for different patterns (relatively to the neighboring peaks) which is also point out on the presence of crystallographic texture.

It seems important to clarify the comments to have the paper ready for publishing.

Author Response

  1. Diffraction patterns on the figure 2 are not indexed.

      Response: Thank you for your remind, we have add index of diffraction patterns in Figure 2 in the revision.

  1. Pure Ga has the only peak on the black pattern. That peak coincides with one of MgGa2. It is not clear why it is interpreted as pure Ga peak instead of crystallographic texture influence on the peak of MgGa2.

      Response: Thank you for your remind, this is a key issue in our article. Firstly, this diffraction peak on the black pattern at 34.996 degree corresponds to Ga (0 2 1) very well; secondly, we observed residual Ga at weld and some areas at fracture surface of 3 s, so we preferably interpreted this peak as Ga.

  1. Intensity of the second peak of Mg2Ga5 is different for different patterns (relatively to the neighboring peaks) which is also point out on the presence of crystallographic texture.

It seems important to clarify the comments to have the article ready for publishing.

Response: Thank you for your professional opinion! We havd been confused about this problem for a long time, but considering the residual Ga observed in fracture surface and weld, although crystallographic texture existed, we prefer to identify this diffraction peak as Ga. And we added the diffraction pattern of 6 s in Figure 2 to provide a comprehensive of changes of the diffraction patterns.

Reviewer 2 Report

Comments and Suggestions for Authors

Review report: Microstructure and properties of magnesium alloy joints bonded by gallium under the assistance of ultrasound at room temperature. Comments are listed below:

1.       Abstract: Add some quantitative results at end of the abstract section.

2.       Introduction: In place of citing multiple references, explain the individual work of the author and try to make a bridge between current and previous work.

3.       Novelty and application: Add a separate section for novelty and application of work.

4.       The purpose of the work should be discussed.

5.       Add the image of the experimental setup. Also mention the purpose of using two different cross-section of the plates.

6.       In XRD analysis, add the quantitative phase results.

7.       Discuss the macrosegragtion at interface areas.

8.       Improve the quality of Fig. 4.

9.       Mention the standard used for sample preparation for shear strength calculation and also image of each specimen extracted along with the image.

10.    Add a separate section related to fracture surface study and also provide a detailed study related to failure mechanism.

11.    In the conclusion section, add key bullet points instead of paragraphs. 

Comments on the Quality of English Language

NA

Author Response

  1. Abstract: Add some quantitative results at end of the abstract section.

      Response: Thank you for your advice! We have added important results at the end of the abstract in the revision.

  1. Introduction: In place of citing multiple references, explain the individual work of the author and try to make a bridge between current and previous work.

      Response: Thank you for your advice! We have expressed references as individual work in the introduction part and made a bridge between them and our work.

  1. Novelty and application: Add a separate section for novelty and application of work.

      Response: Thank you for your advice! We have added a separate section for novelty and application in the final paragraph of introduction: This is the first time to bond magnesium alloy by U-TLP at room temperature without external input of heat source under normal atmosphere.

  1. The purpose of the work should be discussed.

      Response: Thank you for your advice! We have discussed the purpose of the work in the introduction: In this study, to further simplify the device and reduce joining temperature, we joined magnesium alloy using gallium (Ga) as solder under the assistance of ultrasonic vibration.

  1. Add the image of the experimental setup. Also mention the purpose of using two different cross-section of the plates.

      Response: Thank you for your advice! We have add the image of the experimental setup in Figure 1c. The usage of two different cross-section of the plates was to fix the sample on the would s of welding and shear test. The bigger plate was fixed in the would to limit the movement of sample under vibration.

  1. In XRD analysis, add the quantitative phase results.

      Response: Thank you for your advice! We have added some quantitative phase results in the first paragraph in the section of Microstructure of Joint: At 4 s, gallium peak disappeared, a peak of H-MgGa2 at ( 0 1 2 ) appeared at 35.121° and Mg2Ga5 at ( 0 0 2) became the highest peak. This change suggests that at least 4 s ultrasonic time is needed if a reliable joint is expected to obtain. At 6 s, H-MgGa2 at ( 0 1 2) became the highest peak and peak of Mg2Ga5 at ( 0 0 2) declined, which suggested the increase of H-MgGa2 after ultrasonic time of 6 s. At 9 s, the peak of H-MgGa2 declined and Mg2Ga5 became the highest. And we added some connection between this result and other analysis.

  1. Discuss the macrosegragtion at interface areas.

      Response: Thank you for your advice! We think the macrosegragtion was caused by the uneven temperature in different regions at short ultrasonic time and solidification after ultrasound stopped. We added discussion about this in the Effect of Ultrasonic Time on Joint Structure: It may be clear that at short time, the energy provided by ultrasound was not even at different region, thus different structure distributed. For long ultrasonic time, the temperature would be higher and more even, and the high temperature lasted for a longer time, which boosted the generation of more crystal nucleus. Additionally, the fragmentation of IMCs formed preferentially would occurred under the action of ultrasound vibration, so more nucleation points for subsequent grain growth formed [27, 28]. Finally, the good thermal conductivity of Mg base metal means a rapid decrease of temperature when welding finished.

  1. Improve the quality of Fig. 4.

      Response: Thank you for your advice! We have improved the resolution of Figure 4 as well as possible.

  1. Mention the standard used for sample preparation for shear strength calculation and also image of each specimen extracted along with the image.

      Response: Thank you for your advice! We have mentioned the standard used for sample preparation for shear strength in the sections of materials and methods and shear strength of joint.

  1. Add a separate section related to fracture surface study and also provide a detailed study related to failure mechanism.

     Response: Thank you for your advice! We have separated the paragraphs of shear strength and fracture surface, and fracture mechanism has been added in fracture morphology section.

  1. In the conclusion section, add key bullet points instead of paragraphs. 

Response: Thank you for your advice! We have added key bullet points in the conclusion section and added some important points.

(1) AZ31 Mg alloys were successfully jointed using pure Ga at room temperature under the assistance of ultrasound. It is a simple, environment-friendly and safe bonding consideration for low-temperature welding because of the combination of filler with low-melting-point and ultrasound.

(2) The connection of the joint mainly relies on Ga-Mg reaction to generate IMCs (Mg2Ga5, H-MgGa2 and Mg2Ga) and eutectic structure. With the solder Ga gradually consumed, the reliable joint could be obtained after ultrasonic time of 4 s. Afterwards, IMCs would grow and then be refined and homogenized with ultrasonic time further prolonged to 9 s. However, excessive ultrasonic treatment

(3) The joint shear strength increased from 2 s to 4 s and then decreased steady from 4 s to 12 s, and the highest shear strength of 14.65 MPa was obtained after ultrasonic treatment of 4 s. The fracture morphology expressed more characteristics of cleavage fracture with the extension of ultrasonic time.

(4) This study provides a simple and fast welding method for low-temperature joining of magnesium alloy. The mechanical performance indicates it could be used in low-load application at relatively low temperature.

Reviewer 3 Report

Comments and Suggestions for Authors

The manuscript presents a study on the ultrasonic-assisted low-temperature bonding of AZ31 magnesium alloy using gallium solder. The study investigates the microstructure, chemical composition, shear strength, fracture behavior, and the effect of ultrasonic time on these properties. Overall, the study addresses an interesting topic with potential materials science and engineering applications. However, several critical issues need to be addressed before considering this manuscript for publication.

The manuscript needs a comprehensive introduction and background section. It is crucial to provide readers with context regarding the significance of the study, current challenges in magnesium alloy bonding, and how the proposed ultrasonic-assisted method contributes to addressing these challenges.

The literature review needs to be more comprehensive. The authors briefly mention various welding methods for magnesium alloys but need to provide a thorough review of relevant studies in the field. A more comprehensive review would help readers understand the research gap and the novelty of the proposed method.

The manuscript needs more essential experimental details, such as information about the ultrasonic welding system (including frequency, power, and equipment specifications), details of sample preparation, and the specific test conditions for shear strength measurements. These details are necessary to replicate the experiments or assess their validity.

The study provides a range of results, including shear strength values and microstructural observations, yet it needs the inclusion of statistical analyses. Integrating statistical tests would not only bolster the robustness of the findings but also underscore the significance of the results. To illustrate, a more extensive discussion on the growth behavior of IMCs and the processes governing their rate could be undertaken, thereby advancing our comprehension of the kinetics of reactive diffusion at individual interfaces. The authors allude to grain boundary diffusion and rapid diffusion channels, such as dislocations, contributing to IMC formation. It is essential to delve further into this aspect in the discussion section to provide a more comprehensive understanding of these mechanisms.

While the manuscript presents experimental results, the discussion and interpretation of these results are limited. The authors should provide a more in-depth analysis and discussion of the implications of their findings. For example, they should explain why specific microstructures or fracture modes are observed and how they relate to the bonding process.

The conclusion section is relatively brief and lacks a summary of the key findings and their significance. It is essential to summarize the study's main contributions and highlight the practical implications for the field.

Author Response

The manuscript needs a comprehensive introduction and background section. It is crucial to provide readers with context regarding the significance of the study, current challenges in magnesium alloy bonding, and how the proposed ultrasonic-assisted method contributes to addressing these challenges.

Response: As a potential material in areas of structural and functional materials, magnesium alloy is difficult to join by traditional welding methods for its relatively lower melting point and active chemical property, the poor weldability of magnesium and its alloy influenced the universal applications of it. U-TLP bonding is a novel method for magnesium alloy welding, but almost all studies need heating device and the solder used was limited.

The literature review needs to be more comprehensive. The authors briefly mention various welding methods for magnesium alloys but need to provide a thorough review of relevant studies in the field. A more comprehensive review would help readers understand the research gap and the novelty of the proposed method.

Response: Thank you for your suggestion! Because we use the ultrasound as energy provider to bonding, which is similar as U-TLP, we mainly demonstrated the development of U-TLP method in the introduction section. We have divided the conclusive expression of literature of U-TLP to individual work, and improved the quality of introduction as well as possible. Other welding methods were used to introduce the U-TLP, and compared to other U-TLP literature, the most highlight in our work is the usage of liquid solder and the abandon of heating device. We have added some contents about novelty in the last paragraph of introduction section.

The manuscript needs more essential experimental details, such as information about the ultrasonic welding system (including frequency, power, and equipment specifications), details of sample preparation, and the specific test conditions for shear strength measurements. These details are necessary to replicate the experiments or assess their validity.

Response: Thank you for your advice very much! We have added a photo of the ultrasonic welding system in Figure 1, and relative parameters and information of this system were added in the Materials and Methods. Details of sample preparation had been demonstrated as well as possible. We have added the test conditions that we can obtain in the Experimental Process of Materials and Methods. To be honest, most operations of our work at the process of sample preparation and welding were done by hand, so it is simple to express in the literature and errors must exist, but replicating the experiment is easy. We believe a more controllable and precise system would achieve higher and more stable quality of joint.

The study provides a range of results, including shear strength values and microstructural observations, yet it needs the inclusion of statistical analyses. Integrating statistical tests would not only bolster the robustness of the findings but also underscore the significance of the results. To illustrate, a more extensive discussion on the growth behavior of IMCs and the processes governing their rate could be undertaken, thereby advancing our comprehension of the kinetics of reactive diffusion at individual interfaces. The authors allude to grain boundary diffusion and rapid diffusion channels, such as dislocations, contributing to IMC formation. It is essential to delve further into this aspect in the discussion section to provide a more comprehensive understanding of these mechanisms.

Response: Thank you for your professional suggestions again! Because of the limitation of experimental equipment, it is some difficult to collect enough data about diffusion such as the width variation of different layers. Of course, we agree with the idea that a statistic mold will be more convincing and we will built this mold in the following research as try as our best. In the last paragraph of Microstructure of joint, we discussed the formation process of joint

While the manuscript presents experimental results, the discussion and interpretation of these results are limited. The authors should provide a more in-depth analysis and discussion of the implications of their findings. For example, they should explain why specific microstructures or fracture modes are observed and how they relate to the bonding process.

Response: Thank you for your suggestion! We have rearranged some paragraphs and some expression, which might demonstrate the joining of different parts. In general, the ultrasonic time influences the products and microstructure of joint, which determined the fracture location and fracture mode, then influence the shear strength of joint. At room temperature, the reaction of forming IMCs can occur, the introduction of ultrasound fastened this process and changed the size and distribution of IMCs.

In the first part of Results and Discussion, we analyzed the typical microstructure of joint, and demonstrated the formation process of joint. Then at the second part, microstructures of different time were presented, and the effect of ultrasonic time on the diffusion of atoms, growth of grains and refinement and homogenization of IMCs was discussed.in this section. We made some changes of expression in this section to make it more professional and comprehensive. At the final of this section, we implied that microstructure would influence the fracture of joint. Afterwards, in the section of Fraction Locations, we analyzed the locations where fractures generate and extend instead of listing different fracture locations of different ultrasonic time because the organization of joint was not homogeneous. Then in the section of Shear Strength of Joint, we combined the microstructure of joints at different ultrasonic time with fracture locations to explain the reason of the variation of shear strength at different ultrasonic time. In the final section, we gave the fracture morphology at different time, the fracture mode was determined by the composites, microstructure and fracture location of joint.

The conclusion section is relatively brief and lacks a summary of the key findings and their significance. It is essential to summarize the study's main contributions and highlight the practical implications for the field.

Response: Thank you for your advice, we have improved the conclusion section and added key findings and the practical application for the field of our study

(1) AZ31 Mg alloys were successfully jointed using pure Ga at room temperature under the assistance of ultrasound. It is a simple, environment-friendly and safe bonding consideration for low-temperature welding because of the combination of filler with low-melting-point and ultrasound.

(2) The connection of the joint mainly relies on Ga-Mg reaction to generate IMCs (Mg2Ga5, H-MgGa2 and Mg2Ga) and eutectic structure. With the solder Ga gradually consumed, the reliable joint could be obtained after ultrasonic time of 4 s. Afterwards, IMCs would grow and then be refined and homogenized with ultrasonic time further prolonged to 9 s. However, excessive ultrasonic treatment

(3) The joint shear strength increased from 2 s to 4 s and then decreased steady from 4 s to 12 s, and the highest shear strength of 14.65 MPa was obtained after ultrasonic treatment of 4 s. The fracture morphology expressed more characteristics of cleavage fracture with the extension of ultrasonic time.

(4) This study provides a simple and fast welding method for low-temperature joining of magnesium alloy. The mechanical performance indicates it could be used in low-load application at relatively low temperature.

Round 2

Reviewer 1 Report

Comments and Suggestions for Authors

Comments to the manuscript by Qiuyue Fang, Zuoxing Guo, Liang Zhao, Yuhua Liu “Microstructure and properties of magnesium alloy joints bonded by gallium under the assistance of ultrasound at room temperature”

The paper is devoted to the gallium usage as a solder to bond the AZ31 magnesium alloy under ultrasonic assistance at room temperature.

I have the following comments to the corrected paper.

1.      Pure Ga has the only peak on the black pattern (figure 2). One could not identify presence of a specific phase on the basis of just one peak, especially for the low (pure Ga is orthorhombic) symmetry phase.

2.      Intensity of the second peak of Mg2Ga5 is different for different patterns (relatively to the neighboring peaks) which is point out on the presence of crystallographic texture. As soon as presence of crystallographic texture could influence the results it is necessary at least to discuss its presence.

It seems important to clarify the comments to have the paper ready for publishing.

Author Response

Dear reviewer,

Thank you for your time and effort spent this article! Your advice and suggestions help us to improve the quality of this article. We have revised contents according to your advice. New contents were marked as blue, and deleted contents were highlighted as yellow with strikethrough.

Reviewer1

  1. Pure Ga has the only peak on the black pattern (figure 2). One could not identify presence of a specific phase on the basis of just one peak, especially for the low (pure Ga is orthorhombic) symmetry phase.

Response: Thank you for your advice! We would accept your idea that this peak defined as MgGa2, just one peak is not sufficient to identify the existence of a phase. To be honest, the result is not uniform for samples after bonding. Some samples were bonded completely while  in other samples, in some regions of fracture plane, some residual gallium mixed with crystals of IMCs were observed. This inconsistent result confused us for a long time, and led us to make a fault. We have changed the relative contents in the article. Main changes are as following:

  • We changed Figure 2 and the content in abstraction: A reliable jointscomposed of intermetallic compounds (Mg2Ga5, H-MgGa2 and Mg2Ga) and eutectic structure was observed obtained after 4 s ultrasonic treatment of 34 s.

The revised content is as fallowing: A reliable joint composed of intermetallic compounds (Mg2Ga5, H-MgGa2 and Mg2Ga) and eutectic structure was obtained after ultrasonic treatment of 3 s.

  • We changed the contents of XRD analysis from line 175 to line 189: According to the information of PDF cards, at 3 s ultrasonic time, the there was an only diffraction peak of H-MgGa2Gaat ( 0 1 2) was the highest peak, and no peak of Ga appeared, which implies that the solder can be consumed at 3 s. appeared ( 0 2 1) appearing at 34.996°. It implies that gallium was not consumed completely but remained at joint, which would also be demonstrated by the analysis of weld structure and fracture morphology at 3 s. However, when the ultrasonic time prolonged to at 4 s, gallium peak disappeared, a the peak of H-MgGa2 at ( 0 1 2 ) appeared at 35.121° and Mg2Ga5 at ( 0 0 2) became the highest peak. This change suggests that at least 4 s ultrasonic time is needed if a reliable joint is expected to obtain. At 6 s, H-MgGa2 at ( 0 1 2) became the highest peak again and peak of Mg2Ga5 at ( 0 0 2) declined, which suggested the increase of H-MgGa2 after ultrasonic time of 6 s. At 9 s, the peak of H-MgGa2 declined and Mg2Ga5 became the highest. The variation of XRD peaks suggests that the components of IMCs experimented some changes with the increase of ultrasonic time. It is also noteworthy that the shape of diffraction patterns changed at different times, indicating the existence of crystallographic texture.

The revised version is as following: According to the information of PDF cards, at 3 s ultrasonic time, the diffraction peak of H-MgGa2 at ( 0 1 2) was the highest peak, and no peak of Ga appeared, which implies that the solder can be consumed at 3 s. However, at 4 s the peak of Mg2Ga5 at ( 0 0 2) became the highest peak. At 6 s, H-MgGa2 at ( 0 1 2) became the highest peak again and peak of Mg2Ga5 at ( 0 0 2) declined, which suggested the increase of H-MgGa2 after ultrasonic time of 6 s. At 9 s, the peak of H-MgGa2 declined and Mg2Ga5 became the highest. The variation of XRD peaks suggests that the components of IMCs experimented some changes with the increase of ultrasonic time. At 3 s and 6 s, H-MgGa2 was the main component of IMCs, while Mg2Ga5 was the dominant at 4 s and 9 s. It is also noteworthy that the shape of diffraction patterns changed at different times, indicating the existence of crystallographic texture.

  • We revised the expression in results and discussions from line 397 to line 403: When the ultrasonic time increased from 2s to 3s, Ga could be consumed and the amount of IMCs increased.there was only a small amount of residual Ga in the weld, which, different from 2s, was distributed as discontinuous phases in the gaps between IMCs. The fracture mainly occurred in the region composed of Mg2Ga5 and H-MgGa2 and , with very few areas occurring in the residual Ga and some at eutectic layer. Therefore, the average shear strength of the joint increased up to 10.06 MPa.

The revised contents are as following: When ultrasonic time increased from 2s to 3s, Ga could be consumed and the amount of IMCs increased. The fracture mainly occurred in the region composed of Mg2Ga5 and H-MgGa2 and very few occurring at eutectic layer. Therefore, the average shear strength of the joint increased up to 10.06 MPa.

  • In conclusion section, we changed the expression of line 464: With the solder Ga gradually consumed, the reliable joint could be obtained after ultrasonic timeincreasing to of 3 4

The revised version: With the solder Ga gradually consumed, the reliable joint could be obtained after ultrasonic time of 3 s.

  1. Intensity of the second peak of Mg2Ga5 is different for different patterns (relatively to the neighboring peaks) which is point out on the presence of crystallographic texture. As soon as presence of crystallographic texture could influence the results it is necessary at least to discuss its presence.

Response: We appreciate your advice sincerely! We have discussed the presence of crystallographic texture in the first paragraph of Results and Discussions and the analysis of microstructure and shear strength of joint as following:

We added expression as following: It is also noteworthy that the shape of diffraction patterns changed at different times, indicating the existence of crystallographic texture in the analysis of XRD results in line 195 and 196.

Reviewer 2 Report

Comments and Suggestions for Authors

Accepted. 

Comments on the Quality of English Language

NA

Author Response

Dear reviewers,

Thank you for your time and effort spent this article! Your advice and suggestions help us to improve the quality of this article. We have revised contents according to your advice. New contents were marked as blue, and deleted contents were highlighted as yellow with strikethrough.

Reviewer2

Response: Thank you for your advice, we revised some contents and improve English expressions. Your suggestions had helped us to improve the structure of this article.
